# Validation of the Farsi Version of the Adult Concentration Inventory for Assessing Cognitive Disengagement Syndrome

**DOI:** 10.3390/jcm12144607

**Published:** 2023-07-11

**Authors:** Dena Sadeghi-Bahmani, Hadi Parhoon, Maryam Esmaeili, Kamal Parhoon, Laleh Sadeghi Bahmani, Habibolah Khazaie, Stephen P. Becker, G. Leonard Burns, Serge Brand

**Affiliations:** 1Department of Psychology, Stanford University, Stanford, CA 94305, USA; 2Department of Epidemiology & Population Health, Stanford University, Stanford, CA 94305, USA; 3Department of Psychology, Razi University, Kermanshah 6714414971, Iran; h.parhoon@razi.ac.ir; 4Department of Psychology, Faculty of Education and Psychology, University of Isfahan, Isfahan 8174673441, Iran; m.esmaili@edu.ui.ac.ir; 5Department of Psychology, Kharazmi University, Tehran 1571914911, Iran; kamalparhon110@gmail.com; 6Department of Education and Psychology, Shahid Ashrafi Esfahani University, Isfahan 8179949999, Iran; laleh1390sb@gmail.com; 7Sleep Disorders Research Center, Kermanshah University of Medical Sciences (KUMS), Kermanshah 6714415185, Iran; hakhazaie@gmail.com; 8Division of Behavioral Medicine and Clinical Psychology, Cincinnati Children’s Hospital Medical Center, Cincinnati, OH 45229, USA; stephen.becker@cchmc.org; 9Department of Pediatrics, University of Cincinnati College of Medicine, Cincinnati, OH 45267, USA; 10Department of Psychology, Washington State University, Pullman, WA 99164, USA; glburns@wsu.edu; 11Center for Affective, Sleep and Stress Disorders, Psychiatric Clinics of the University of Basel, 4002 Basel, Switzerland; serge.brand@upk.ch; 12Division of Sport Science and Psychosocial Health, Department of Sport, Exercise and Health, Faculty of Medicine, University of Basel, 4052 Basel, Switzerland; 13Substance Abuse Prevention Research Center, Kermanshah University of Medical Sciences (KUMS), Kermanshah 6714415185, Iran; 14School of Medicine, Tehran University of Medical Sciences (TUMS), Tehran 1441987566, Iran; 15Center for Disaster Psychiatry and Disaster Psychology, Psychiatric Clinics of the University of Basel, 4002 Basel, Switzerland

**Keywords:** cognitive disengagement syndrome, sluggish cognitive tempo, Farsi, psychometrics, confirmatory factor analysis

## Abstract

The internal and external validity of cognitive disengagement syndrome (CDS) relative to attention-deficit/hyperactivity disorder-inattention (ADHD-IN) was evaluated herein with Farsi-speaking adults. A total of 837 Iranian adults assessed throughout the whole country (54.72% women, M_age_ = 23.85; SD = 7.05; age range = 18 to 58 years; 75% between 18 and 24 years old; reporting higher educational training) completed self-report measures of CDS, ADHD-IN, ADHD-hyperactivity/impulsivity (HI), depression, anxiety, and stress. Seven of the fifteen CDS symptoms showed a good convergent (high loadings on the CDS factor) and discriminant (higher loadings on the CDS factor than the ADHD-IN factor) validity. CDS also showed stronger first-order and unique associations with depression than ADHD-IN, whereas ADHD-IN showed stronger first-order and unique associations with ADHD-HI and anxiety than CDS. The first-order and unique associations of CDS and ADHD-IN did not differ in relation to stress. This study is the first to support the validity of the self-report of assessing CDS symptoms with the Adult Concentration Inventory with Farsi-speaking individuals residing in Iran, thus further strengthening the transcultural validity of the CDS, and paving the way for further transcultural research in the field of CDS among adults.

## 1. Introduction

In recent years, a growing body of research has focused on cognitive disengagement syndrome (CDS), formerly sluggish cognitive tempo (SCT), which refers to a set of attentional symptoms that includes daydreaming, staring, mental fogginess/confusion, and slowed behavior/thinking [1]. Note that we use the term cognitive disengagement syndrome (CDS) throughout the present paper, while the publications reported here may have used the term sluggish cognitive tempo (SCT). Cross-sectional, longitudinal, and meta-analytic studies have consistently reported the robust pattern of CDS being distinguished from inattentive symptoms of the attention-deficit/hyperactivity disorder (ADHD-IN) [2]. Importantly, among children and adolescents, such a pattern of CDS could be observed among different cultural areas and languages such as American English [3,4,5,6], Spanish [7,8], Korean [9], Turkish [10,11], and Farsi [12]. Thus, while it appears that the concept of CDS is well-established among children and adolescents, this is less the case for adults in general, and more specifically for Farsi-speaking adults. 

As regards the internal and external validity of 15 frequently-used CDS items for children and adolescents, the parent and teacher rating-scale studies from a variety of countries have shown satisfactory psychometric properties, in that the 15 CDS items loaded differently on the CDS and ADHD-IN factors (Iran: Ref [12]; South Korea: Ref [9]; Spain: Ref [8,13]; Turkey: Ref [10]; United States: Ref [5,14]).

In contrast, studies with self-report scales in adolescents and adults have shown that not all of the 15 CDS items had discriminant validity with the ADHD-IN factor; that is to say that certain CDS items had equal loadings on both the CDS and ADHD-IN factors, and thus an unsatisfactory discriminant validity [15,16,17,18,19,20,21]. Given this, the present study is a continuation of this research endeavor to identify the self-rated CDS symptoms with a good internal validity relative to the ADHD-IN factor (i.e., higher loadings on the CDS factor than the ADHD-IN factor) and to examine the external validity of the CDS factor relative to the ADHD-IN. The present study is also the first to examine this question with adults from Iran.

### 1.1. Adult Symptoms of Cognitive Disengagement Syndrome (CDS) in Relation to Cognitive, Attentional, Emotional, Behavioral, and Sleep-Related Difficulties

Data from 60 cross-sectional and longitudinal studies have shown that higher scores for CDS are associated with a poorer academic performance [22]. More specifically, higher CDS symptoms have been associated with: (i) poorer skills in daily life executive functioning and a greater functional impairment in the specific domains of educational activities, work, money/financial issues, managing chores and household tasks, community activities, and social situations with strangers and friends [23]; (ii) lower scores for self-organization and problem solving; (iii) higher scores for a distorted time perception [18]; (iv) a weaker orienting network due to the problems of engaging and disengaging attention [24]; (v) a reduced speed and efficacy of selective attention in early information processing [25]; (vi) more deficits in the use of self-regulated learning strategies [26]; and (vii) more difficulties in a timed reading test, although students with CDS were not slower than controls in reading comprehension, processing speed, and reading fluency [27]. On this point, more recent research has provided clearer evidence for a link between CDS and a wide range of academic [28] and neurocognitive [29] outcomes.

As regards studies on CDS focusing on emotional difficulties, including stress and social behavior, adults scoring high on CDS have been shown to be at an increased risk of reporting symptoms of suicidal behavior, beyond other health dimensions, including symptoms of depression [16]. Next, higher scores for ADHD have been shown to predict higher scores for self-perceived stress, while the combination of symptoms of inattentive-type ADHD (ADHD-IN) and symptoms of CDS represents the most consistent predictor of perceived stress [30]. Higher scores for emotion dysregulation have been shown to moderate the associations between higher scores for CDS and social impairment. Importantly, CDS traits and social withdrawal appear to be highly intertwined [31]. To explain such an association, the conceptual framework considers task-unrelated thoughts, poorer social skills, and social anxiety, along with possible moderators such as behavioral inhibition and unfavorable parenting styles, in the emergence and maintenance of the link between CDS and social withdrawal [32]. 

As regards the associations between CDS scores and sleep parameters, higher scores for CDS have been associated with more impaired sleep patterns [33,34] and lower sleep quality, along with more daytime dysfunction [35], more sleep disturbances, and higher scores for symptoms of stress, depression, and functional impairment [36]. 

### 1.2. Assessing Cognitive Disengagement Syndrome among Adults

In the meantime, measures to assess CDS traits among adults have emerged and been psychometrically validated. Two measures have been used primarily: the Barkley Adult ADHD Rating Scale-IV (BAARS-IV; Sluggish Cognitive Tempo subscale [37]; four studies: Refs [38,39,40,41]) or the Adult Concentration Inventory [15,17,18]. For the English BAARS-IV, a three-factor solution was observed for the American-English version [39,40]; a two-factor solution was observed for the Turkish version [41], and a four-factor solution was observed for the Japanese version [38]. Further, to our knowledge, the American-English ACI [15,17] has only been translated into Brazilian-Portuguese [18,19].

### 1.3. The Present Study

The purpose of the study was to evaluate for the first time the construct validity of scores from the 15-item CDS self-report ACI scale with Iranian adults.

The first objective was to determine the internal (structural) validity of the 15 CDS symptoms with the 9 ADHD-IN symptoms. As noted earlier, these 15 CDS symptoms have demonstrated convergent and discriminant validity with the ADHD-IN symptoms from parent and teacher ratings of the 15 CDS and 9 ADHD symptoms obtained from numerous countries including Iran; however, the CDS symptoms with convergent and discriminant validity relative to the ADHD-IN symptoms within the self-report measure have been inconsistent across studies. The first objective was therefore to further attempt to identify the best CDS symptoms on the self-report measures. The identification of CDS symptoms with a good internal validity relative to ADHD-IN symptoms on the self-report measures would then allow for the determination of the external correlates of the CDS and ADHD-IN symptom dimensions. 

The second objective was to evaluate the invariance of the CDS symptoms across men and women. It was expected that the CDS symptoms would show an invariance of like-item loadings and thresholds across men and women with no significant difference in the CDS factor mean. Given that the studies on the 15 CDS symptoms with the parent and teacher rating scales found an invariance of like-item loadings and thresholds across boys and girls [6,8,10,14], we hypothesized that the same results would occur across men and women.

The third and fourth objectives evaluated the external validity of the CDS and ADHD-IN factors with other symptom factors (i.e., ADHD-HI, depression, anxiety, and stress). It was expected that CDS would have a stronger correlation and unique association (partial regression coefficient) with depression compared with ADHD-IN, whereas ADHD-IN would have a stronger correlation and unique association than CDS with ADHD-HI [6,8,9,10,12,13,14,15]. It was also expected based on earlier studies that the unique association of CDS with ADHD-HI would be negative in the regression analysis (i.e., higher scores on CDS would be associated with lower scores on ADHD-HI after controlling for the overlap of CDS and ADHD-IN). For anxiety and stress, given the high overlap between symptoms of depression, anxiety, and stress, at least among clinical samples other than individuals with CDS and ADHD [42,43,44], we expected that CDS would have a stronger correlation and unique association (partial regression coefficient) with anxiety and stress than ADHD-IN.

We hold that for the following three reasons, the present validation study is important. First, Farsi is the official language of Iran, Afghanistan, and Tajikistan [45]. Further, it is estimated that about 77.4 million people speak Farsi as their first language, and 40 million people use Farsi as their second language [46], in addition to minorities in Uzbekistan, the United States of America, Pakistan, Turkey, United Arab Emirates, Iraq, Qatar, Germany, India, or Canada [45]. Second, the validation of the Farsi ACI should allow for the corroboration of the prevalence and stability of CDS among adults, irrespective of cultural background. Third, several studies among non-adult samples with CDS have shown that children and adolescents scoring high on CDS might respond differently to evidence-based treatments for ADHD [47,48,49]. As such, the Farsi version of the ACI might help to identify those adults in need of a more appropriate or monitored psychopharmacological and psychotherapeutic treatment.

## 2. Materials and Methods

### 2.1. Participants

Participants were recruited from provinces of each region of Iran including the north (Mazandaran), south (Shiraz), east (Kermanshah), west (Mashhad), and center (Tehran and Esfahan). Participants were recruited via advertisements on the intranet of universities, private companies, and sports and leisure time activity associations, including associations for people in retirement. The inclusion criteria were: (i) age between 18 and 65 years; (ii) willing and able to complete a series of self-rating questionnaires in Farsi; (iii) signed written informed consent. The exclusion criteria were: (i) current self-reported psychological, psychiatric, and/or somatic issues, which might have biased the results of the questionnaires; (ii) pregnant or breast-feeding women, as these two conditions might modify the current cognitive-emotional processing. A total of 868 people completed the questionnaire with 31 participants (3.57%) excluded for self-reported current psychological, psychiatric, or somatic issues. The final sample consisted of 837 individuals with a mean age of 23.58 years (*SD* 7.05; age range: 18 years to 58 years). Of these, 75% were between 18 and 24 years old; 16% were between 25 and 35 years, and 9% were between 36 and 58 years old, with 54.7% female and 24% married. Educational degrees were reported: diploma (33.9%), bachelor’s degree (53.6%), master’s degree (9.1%), and doctoral degree (3.3%).

### 2.2. Procedures

The ethics committee of the Kermanshah University of Medical Sciences (Kermanshah, Iran) approved the study (ethical code: IR.KUMS.REC.1398.016), which was performed in accordance with the seventh and current revision [50] of the Declaration of Helsinki.

To accurately translate the English version of the ACI into Farsi, we followed the algorithms proposed by Brislin [51], Beaton, et al. [52], and Sousa and Rojjanasrirat [53] (see also [12]): (i) two independent translators with expertise in both Farsi and English translated the English version of the ACI into Farsi; (ii) a third independent person with expertise in both Farsi and English compared the two translations; (iii) where there were differences, the three experts discussed the issues and formulated the final draft; (iv) two further and new independent translators with expertise in both Farsi and English performed the back-translation, and (v) compared the back-translated English version with the original version; (vi) the final version reflected the general agreement of all five researchers involved in this procedure. The Farsi version of the Adult Concentration Inventory is available in Appendix A. Please note that to maintain objectivity and independence as much as possible, none of the present authors and co-authors were involved in the translation process. 

### 2.3. Measures

#### 2.3.1. Adult Concentration Inventory (ACI)

The ACI is a self-report measure of 16 CDS symptoms, although one item assessing low motivation has consistently shown a poor convergent and discriminant validity [17] and was not used in this study. Each item is rated on a four-point scale (0 = not at all, 1 = sometimes, 2 = often, and 3 = very often). As regards the internal and external validity of these 15 CDS items for children and adolescents, the parent and teacher rating-scale studies from a variety of countries have shown satisfactory psychometric properties, in that the 15 CDS items loaded differently on the CDS and ADHD-IN factors (Iran: Ref. [12]; South Korea: Ref. [9]; Spain: Refs. [8,13]; Turkey: Ref. [10]; United States: Refs. [5,14]). In contrast, the psychometric properties of the self-report scales of the 15 CDS items have shown that that not all of the 15 CDS self-report items have shown discriminant validity with the ADHD-IN factor; that is to say that certain CDS items had equal loadings on both the CDS and ADHD-IN factors, and thus an unsatisfactory discriminant validity [15,16,17,18,19,20].

#### 2.3.2. Barkley Adult ADHD Rating Scale-IV (BAARS-IV, Barkley, 2011 [37])

The Farsi version [54] of the BAARS-IV [37] was used to obtain self-report measures of the nine ADHD-IN and nine ADHD-HI symptoms. Each item was rated on a four-point scale (0 = not at all, 1 = sometimes, 2 = often, and 3 = very often). For the current sample, the Cronbach’s alpha values for the ADHD-IN and ADHD-HI scales were 0.84 and 0.80. The ADHD-IN and ADHD-HI scores from the BAARS-IV have demonstrated satisfactory psychometric properties in many studies across various countries [37,38,39,40,41].

#### 2.3.3. Depression Anxiety Stress Scale (DASS-21, Lovibond and Lovibond, 1995 [55])

The Farsi version [56] of the DASS-21 self-report scale [55] was used to measure depression (seven items, e.g., *I felt down hearted and blue*), anxiety (seven items, e.g., *I experienced trembling in my hands*), and stress (seven items, e.g., *I found it difficult to relax*). Each item was rated on a four-point scale (i.e., 0 = never, 1 = sometimes, 3 = often, and 4 = almost always). For the current sample, the Cronbach’s alpha values for the depression, anxiety, and stress scales were 0.87, 0.82, and 0.83, respectively. Scores from the depression, anxiety, and stress scales have shown positive psychometrics in earlier research [57,58].

### 2.4. Analytic Strategy 

The Mplus statistical software was used for the analyses (Version 8.8, [59]). The self-report items were treated as categorical indicators with the robust weighted least squares estimator used for the analyses (WLSMV). The global model fit was evaluated with the comparative fit index (CFI, acceptable fit ≥ 0.90, close fit ≥ 0.95), the root-mean-square error of approximation (RMSEA, acceptable fit ≤ 0.08, close fit < 0.05), and the standardized root-mean-square residual (SRMR, acceptable fit ≤ 0.08, close fit ≤ 0.05; Little, 2013). 

The first analysis applied an a priori CDS and ADHD-IN two-factor model to the 15 CDS items and the 9 ADHD-IN items. Items were allowed to have cross-loadings (i.e., exploratory confirmatory factor analysis, as per Asparouhov and Muthén [60]). The purpose of this analysis was to identify CDS items with high loadings on the CDS factor (convergent validity) and higher loadings on the CDS factor than the ADHD-IN factor (discriminant validity). CDS items were required to load higher than 0.50 on the SCT factor and lower than 0.30 on the ADHD-IN factor to be retained as indicators of the CDS construct.

The second analysis determined the correlations of the CDS and ADHD-IN factors with the ADHD-HI, depression, anxiety, and stress factors. For this six-factor model, the CDS items were allowed to cross-load on the ADHD-IN factor and the ADHD-IN items to cross-load on the CDS factor. No other cross-loadings were allowed in this factor model [60]. The purpose of this analysis was to determine the correlations of the CDS and ADHD-IN factors with the ADHD-HI, depression, anxiety, and stress factors. The Mplus model constraint procedure was used to test for significant differences in the associations of the CDS and ADHD-IN factors with the other factors.

The third analysis regressed the ADHD-HI, depression, anxiety, and stress factors on the CDS and ADHD-IN factors. This analysis also allowed the CDS items to cross-load on the ADHD-IN factor and the ADHD-IN items to cross-load on the CDS factor with no other cross-loadings allowed in the model (i.e., exploratory structural regression model; Asparouhov and Muthén [60]). The purpose of this analysis was to determine the unique associations of the CDS and ADHD-IN factors with the ADHD-HI, depression, anxiety, and stress factors. The model constraint procedure was used to determine if the partial unstandardized regression coefficients from the regression of the ADHD-HI, depression, anxiety, and stress factors on the CDS and ADHD-IN factors differed in strength.

## 3. Results

### 3.1. Internal Validity of CDS and ADHD-IN Symptoms 

Table 1 shows the results from the application of an a priori CDS and ADHD-IN two-factor model. In total, 7 of the 15 CDS symptoms met our criteria for convergent validity (i.e., loadings greater than 0.50 on the CDS factor) and discriminant validity (i.e., loadings less than 0.30 on the ADHD-IN factor). These seven CDS items were: (1) *I am slow at doing things*; (2) *My mind feels like it is in a fog*; (4) *I feel sleepy or drowsy during the day*; (7) *I get lost in my own thoughts*; (8) *I get tired easily*; (9) *I forget what I was going to say*; and (11) *I zone or space out*. These seven items were retained to define the CDS construct. Cronbach’s alpha for these items was 0.83.

A CDS and ADHD-IN factor model applied to the seven CDS and nine ADHD-IN items with cross-loadings yielded a close fit: *χ*^2^ (89) = 266, *p* < 0.001, CFI = 0.979, RMSEA = 0.049 (0.042, 0.056), and SRMR = 0.028. The average loading of the seven CDS items on the CDS factor was 0.64 (*SD* = 0.11), with the average loading of the CDS items on the ADHD-factor being 0.08 (*SD* = 0.12). The average loading of the nine ADHD-IN symptoms on the ADHD-IN factor was 0.67 (*SD* = 0.10), with the average loading on the CDS factor being 0.02 (*SD* = 0.13). The seven CDS items thus showed a good convergent and discriminant validity with the nine ADHD-IN items, also showing a good convergent and discriminant validity for the sample. The correlation between the CDS and ADHD-IN factors was 0.69. The CDS and ADHD-IN factors thus maintained 52% of their true score variance independent of the other factor.

### 3.2. Invariance of CDS Items across Sex

A one-factor model with constraints on like-item loadings and like-item thresholds across men and women yielded an acceptable fit: *χ*^2^ (48) = 99, *p* < 0.001, CFI = 0.987, RMSEA = 0.051 (0.036, 0.065), and SRMR = 0.037. The model with the constraints on like-item loadings and like-item thresholds did not result in a significant decrement in fit relative to the model without the constraints: *χ*^2^ (20) = 18.89, *p* = 0.45. Although women had a slightly higher CDS factor mean than men, the difference was not significant: Cohen’s latent *d* = 0.08, *SE* = 0.08, *p* = 0.29.

### 3.3. Correlations of CDS and ADHD-IN Factors with ADHD-HI, Depression, Anxiety, and Stress Factors

Table 2 shows the correlations of the CDS and ADHD-IN factors with the ADHD-HI, depression, anxiety, and stress factors. This model yielded an acceptable fit: *χ*^2^ (960), 3063, *p* < 0.001, CFI = 0.928, RMSEA = 0.051 (0.049, 0.053), and SRMR = 0.055. The CDS factor had a stronger correlation than the ADHD-IN factor with the depression factor (*p* < 0.001), whereas the ADHD-IN factor had a stronger correlation than the CDS factor with the ADHD-HI and anxiety factors (*p* < 0.001). The CDS and ADHD-IN factors did not differ in their association with the stress factor (*p* > 0.10).

### 3.4. Unique Associations of CDS and ADHD-IN Factors with ADHD-HI, Depression, Anxiety, and Stress Factors

Table 3 shows the partial standardized regression coefficients from the regression of the ADHD-HI, depression, anxiety, and stress factors on the CDS and ADHD-IN factors. The fit of this mode was the same as the previous model. The CDS factor had a stronger unique association with the depression factor (*p* < 0.001) than the ADHD-IN factor, whereas the ADHD-IN factor had a stronger unique association than the CDS factor with the ADHD-HI (*p* < 0.001) and anxiety (*p* < 0.01) factors. The CDS and ADHD-IN factors did not differ significantly in their unique associations with the stress factor (*p* > 0.10). In addition, the unique associations of the CDS and ADHD-IN factors with the other symptom factors remained the same after controlling for participants’ age, sex, and marital status.

## 4. Discussion

The purpose of the study was to evaluate for the first time the construct validity of scores from the 15-item CDS self-report scale with Iranian adults. More specifically, four objectives were formulated. The first objective was to determine the internal (structural) validity of the 15 CDS symptoms with the 9 ADHD-IN symptoms. The second objective was to evaluate the invariance of the CDS symptoms across men and women, and the third and fourth objectives evaluated the external validity of the CDS and ADHD-IN factors with other symptom factors (i.e., ADHD-HI, depression, anxiety, and stress). Results evidenced that 7 of the 15 CDS symptoms showed a good convergent (high loadings on the CDS factor) and discriminant (higher loadings on the CDS factor than the ADHD-IN factor) validity. These seven CDS symptoms showed an invariance of like-item loadings and thresholds across men and women, with no significant difference in the CDS factor mean. Further, CDS also showed stronger first-order and unique associations than ADHD-IN with depression, whereas ADHD-IN showed stronger first-order and unique associations than CDS with ADHD-HI and anxiety. The first-order and unique associations of CDS and ADHD-IN did not differ with stress.

The current study is the first to support the validity of the self-report of CDS symptoms with adults from Iran, thus further strengthening the transcultural validity of CDS [61]. In this view, in our opinion, the present results add to the current literature in three important ways. First, the Farsi version of the ACI [15,21] has highly satisfactory psychometric properties. Second, to our knowledge, this is only the second non-English validation of the ACI (Brazilian-Portuguese version: Refs. [18,19]); former translations used the Barkley Adult ADHD Rating Scale-IV (BAARS-IV) [37] to assess adult CDS traits (Japanese: Ref. [38]; Turkish: Ref. [41]). Third, CDS traits correlated with dimensions of depression, anxiety, and stress, thus confirming that CDS traits and internalizing symptoms seem to appear concomitantly. 

### 4.1. Internal (Structural) Validity of the 15 CDS Items with the 9 ADHD-IN Symptoms

The results showed (Table 1) that a seven-item solution yielded the best statistical fit for the internal (structural) validity of the 15 CDS items. Previous studies with US-American samples [15,17,20] and a Brazilian sample [18] observed a ten-item solution. Importantly, the observed seven-item solution helped to distinguish CDS from ADHD-IN and allowed, in a further step, the determination of the external correlates of the CDS and ADHD-IN symptom dimensions. Using the Child and Adolescent Behavior Inventory (CABI) [62] to assess the dimensions of CDS/SCT among children and adolescents (parent ratings), the Farsi version [12] and the Korean version [9] yielded 11 out of 15 items; the Spanish version [13] yielded 15 out of 16 items; and the Turkish version [11] yielded 12 out of 15 items. 

Questions might arise as to what extent the identification of 7 out of 15 items might be considered a satisfactory pattern of results, when compared to the ratio of 9 out of 16 items [20], or 10 out of 16 items [15,17] for the ACI. To our knowledge, there is no well-defined and definite cut-off value; we claim that the seven identified items yielded satisfactory psychometric properties, including both a statistically significant convergent and divergent construct validity, with an identical pattern of results for both female and male participants, and with stronger first-order and unique associations for CDS than ADHD-IN with depression, whereas ADHD-IN showed stronger first-order and unique associations than CDS with ADHD-HI and anxiety. 

### 4.2. Invariance of CDS Scores across Men and Women

The seven CDS symptoms showed an invariance of like-item loadings and thresholds across men and women with no significant difference in the CDS factor mean. In our opinion, this result is important for the following reasons. While prevalence rates and possible gender differences in CDS among children, adolescents, and adults are modest at most [3], it is important that the scientific community can rely on a psychometrically sound and valid measure to assess adult CDS for both male and female adults. From a broad range of studies on the relation between adult CDS and cognitive, attentional, emotional, behavioral, and sleep-related difficulties (for details, see the Introduction), none have identified specific gender-related patterns. As such, the present Farsi version of the ACI to assess CDS among adults is a valid and reliable measure for both female and male adults. 

### 4.3. Cognitive Disengagement Syndrome and Internalizing Problems

The overall pattern of results showed that higher scores for CDS were associated with higher scores for symptoms of depression. As such, the present data corroborated what has already been observed [16,38,39,63], thus strengthening the quality and reliability of the present data.

As regards the observed associations between higher scores for CDS and higher scores for stress, it appears that this is only the second study in this field. Among 983 adults aged on average 45.6 years, higher scores for ADHD predicted higher scores for self-perceived stress, while the combination of symptoms of ADHD-IN and symptoms of CDS was the most consistent predictor of perceived stress [30]. The present data thus strengthen the sparse evidence on the relation between CDS and self-perceived stress. 

Next, the associations between CDS and anxiety demand particular attention. Higher scores for ADHD-IN were associated more strongly than scores for CDS with higher scores for anxiety, which appears to contrast with previous studies on CDS and anxiety [15,38,39,63]. Such a pattern appears to suggest that CDS is unrelated, or at least less related to anxiety, than scores for ADHD-IN. However, we hold that such an interpretation should consider the following observations. First, symptoms of depression and anxiety are highly related [64], at least among clinical samples broader than individuals with CDS and ADHD [42,43,44]. Second, the so-called transdiagnostic approach [65,66,67,68] considers mental health issues within a spectrum ranging from symptoms of depression to symptoms of anxiety, and applying the umbrella term of internalizing problems. Relatedly and third, in previous research, higher scores for CDS were associated with higher scores for internalizing problems, including poorer emotion regulation [31], daily life deficits [15], issues in socio-emotional adjustment such as anxiety, depression, loneliness, emotion dysregulation, low self-esteem, global functional impairment [3], and academic [40] and general adjustment difficulties and impairments [32]. Given this, we also claim that the present pattern of results suggests that adults scoring high on CDS also score high on internalizing problems and emotion dysregulation, in general, and on symptoms of anxiety, more specifically, even if this occurs in the present study to a lesser degree than ADHD-IN symptoms.

### 4.4. Limitations

The present validation study has the following limitations. First, previous studies have shown that (adult) CDS traits are associated with more sleep difficulties [33,34,36]; thus, future studies among (Iranian) adults should consider this health issue. This is even more justified in that there are proven and standardized psychotherapeutic interventions to improve sleep, such as: (i) insomnia-specific CBT interventions, either in vivo [69,70,71,72,73,74,75,76] or via online tools (CBT-i: Refs. [77,78]); (ii) acceptance and commitment therapy (ACT) [79]; (iii) sleep hygiene interventions among adolescents [80,81,82], just to name but three standardized cognitive-behavioral interventions. Further, to illustrate, and at least among children aged about 10 years with diagnosed ADHD, thorough sleep hygiene training involving above all the children’s parents and lasting for 12 consecutive weeks led to remarkable improvements in the children’s sleep quality, ADHD-related behavior, and school performance [83]. Similarly, we assume that adults reporting both high scores for CDS and for sleep difficulties might benefit from a thorough and standardized sleep-intervention program to improve both their sleep issues and everyday issues related to CDS traits.

Second, we assessed an explicitly non-clinical sample of typically developed adults consisting prevalently of participants in their young adulthood and with a solid higher educational background. In contrast, while the prevalence rate of CDS among adults appears to be unclear, the prevalence of ADHD among adults ranges between 0.23% and 2.18% among older adults [84], and between 2.58% and 6.76% among the general adult population [85,86]. As such, we must assume that future studies on adult CDS without limiting inclusion and exclusion criteria should provide more information as regards the associations between adult CDS traits, vocational/academic achievements, symptoms of psychopathology including non-suicidal self-injury and suicidal behavior, and daily executive functioning, including the quality of social relationships with peers, family members, romantic partners, or colleagues at work and in leisure time.

## 5. Conclusions

The Farsi version of the Adult Concentration Inventory (ACI) to identify CDS traits among adults is a valid and psychometrically sound measure. Further, the measure might allow for a further and more thorough exploration as to whether adults with ADHD might benefit from a deeper investigation of CDS, particularly in determining treatment routes. Froehlich, Becker, Nick, Brinkman, Stein, Peugh, and Epstein [48] have shown, among a smaller sample of 126 stimulant-naïve children with ADHD-IN and 45 stimulant-naïve children with ADHD-IN and ADHD-HI, that non-responders to methylphenidate are those participants scoring high on CDS, including daydreaming and daytime sleepiness. As such, it is conceivable that a similar pattern might also be observed among adults.

## Figures and Tables

**Table 1 jcm-12-04607-t001:** Standardized primary and secondary factor loadings (SEs) of cognitive disengagement syndrome and ADHD-inattention symptoms on cognitive disengagement syndrome and ADHD-inattention factors.

Cognitive Disengagement Syndrome Symptoms	CDS Factor	ADHD-IN Factor
**1. I am slow at doing things**	0.57 (0.06)	0.04 (0.07) ^ns^
**2. My mind feels like it is in a fog**	0.60 (0.07)	0.10 (0.08) ^ns^
3. I stare off into space	0.39 (0.06)	0.20 (0.07)
**4. I feel sleepy or drowsy during the day**	0.79 (0.06)	−0.10 (0.08) ^ns^
5. I lose my train of thought	0.37 (0.05)	0.00 (0.04) ^ns^
6. I am not very active	0.37 (0.07)	0.38 (0.07)
**7. I get lost in my own thoughts**	0.78 (0.06)	−0.09 (0.08) ^ns^
**8. I get tired easily**	0.52 (0.07)	0.27 (0.07)
**9. I forget what I was going to say**	0.63 (0.06)	0.07 (0.08) ^ns^
10. I feel confused	0.17 (0.07)	0.53 (0.06)
**11. I zone or space out**	0.75 (0.05)	0.02 (0.07) ^ns^
12. My mind gets mixed up	0.36 (0.07)	0.34 (0.07)
13. My thinking seems slow or slowed down	0.18 (0.07)	0.50 (0.06)
14. I daydream	0.22 (0.07)	0.47 (0.07)
15. I have a hard time putting my thoughts into words	0.39 (0.06)	0.19 (0.07)
**ADHD-Inattention Symptoms**		
1. Close attention to detail	0.01 (0.06) ^ns^	0.65 (0.05)
2. Sustaining attention	0.09 (0.07) ^ns^	0.63 (0.06)
3. Listening when spoken to directly	−0.06 (0.07) ^ns^	0.72 (0.06)
4. Follow through on instructions	0.00 (0.01) ^ns^	0.73 (0.03)
5. Organization skills	0.05 (0.07) ^ns^	0.70 (0.06)
6. Avoids tasks requiring sustained effort	0.15 (0.07)	0.49 (0.06)
7. Loses things	−0.27 (0.07)	0.81 (0.06)
8. Easily distracted	0.29 (0.07)	0.54 (0.06)
9. Forgetful	−0.08 (0.07) ^ns^	0.76 (0.06)

Note. *N* = 837. The bold CDS symptoms met our convergent and discriminant validity criteria (i.e., a loading greater than 0.50 on the CDS factor and a loading less than 0.30 on the ADHD-IN factor). These seven CDS symptoms were used in the analyses shown in Table 1 and Table 2. All loadings were significant at *p* < 0.05 unless noted as non-significant (ns). The complete wording of the ADHD-IN symptoms is shown on the BAARS. ADHD = attention-deficit/hyperactivity disorder; IN = inattention; CDS = cognitive disengagement syndrome.

**Table 2 jcm-12-04607-t002:** Correlations (SEs) of the CDS and ADHD-IN factors with other factors.

External Correlates	CDS	ADHD-IN
ADHD-HI	0.28 (0.04) ^a^	0.65 (0.03) ^b^
Depression	0.82 (0.02) ^a^	0.66 (0.03) ^b^
Anxiety	0.58 (0.03) ^a^	0.69 (0.03) ^b^
Stress	0.72 (0.02) ^a^	0.70 (0.03) ^a^

Note. *N* = 837. All correlations were significant at *p* < 0.001. Correlations for the same external correlate with different superscripts differ at *p* < 0.001. CDS = cognitive disengagement syndrome; ADHD = attention-deficit/hyperactivity disorder; IN = inattention; HI = hyperactivity/impulsivity.

**Table 3 jcm-12-04607-t003:** Partial standardized regression coefficients (SEs) for the association of CDS and ADHD-IN factors with other symptom factors.

External Correlates	CDS	ADHD-IN
ADHD-HI	−0.28 (0.07) ^a^	0.84 (0.06) ^b^
Depression	0.68 (0.04) ^a^	0.21 (0.05) ^b^
Anxiety	0.21 (0.05) ^a^	0.55 (0.05) ^b^
Stress	0.46 (0.05) ^a^	0.40 (0.05) ^a^

Note. *N* = 837. All partial standardized regression coefficients were significant at *p* < 0.001. Regression coefficients for the same external correlate with different superscripts differ at *p* < 0.01. The significance tests were performed on the partial unstandardized regression coefficients. The results also remained the same controlling for age, sex, and marital status. CDS = cognitive disengagement syndrome; ADHD = attention-deficit/hyperactivity disorder; IN = inattention; HI = hyperactivity/impulsivity.

## Data Availability

Data are made available to explicit experts in the field. Such experts should clearly formulate their hypotheses; further, they should fully describe how and where they will securely store the data file and how they will ensure that the data file is not shared with/is securely protected from third parties.

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
