# Peer review of "Validation of the Farsi Version of the Adult Concentration Inventory for Assessing Cognitive Disengagement Syndrome"

_jcm, 2023, doi:10.3390/jcm12144607_

Round 1
Reviewer 1 Report
I find the manuscript to be well-written in general. However, I would like to offer some comments and suggestions for improvement.
- I would suggest that the author includes the expanded form of "CDS" (Cognitive Disengagement Syndrome) in the abstract to avoid any ambiguity. This will provide clarity to readers, especially since "CDS" can also stand for "Concomitant Difficulty Scale" in the context of ADHD.
- In your study, it was found that 7 out of the 15 CDS items demonstrated good convergent validity. The author noted that it would have been valuable to report how the 15 items of the CDS scale loaded differently on both the CDS and ADHD-IN factors in various countries such as Iran, South Korea, Spain, Turkey, and the United States. Providing information on the number of items with good convergent validity in these countries, including the original CDS scale if applicable, would have enhanced the usefulness of the findings.
- In the introduction section the author mentioned, some studies showed "unsatisfactory discriminant validiy". How would you categorize the discriminant validity in this study, satisfactory or unsatisfactory?
- In the last paragraph of introduction you mentioned that Persian is the official language of 3 coutnries. The following statements narrate how widely Farsi is spoken. Are Persian and Farsi the same language? As I read through the manuscript, it became clearer that both are the same languages, but initially it's cnofusing.
- Information in the 2nd and 3rd paragraphs of introduction section is repeated in the section 2.3.1. Adult Concentration Inventory section of Materials and Methods except citations.
- In section 4.2, CDS is misspelled as DCS.
Author Response
Dear Reviewer,
Thank you very much for all your kind efforts.
We have addressed all concerns raised by the Reviewers. Please see the detailed point-by-point-response attached as a separate file.
Again, thank you very much for the care devoted to our manuscript.

Reviewer 2 Report
The article "Validation of the Persian Version of the Adult Concentration Inventory for Assessing Cognitive Disengagement Syndrome" is a report about the validation of ADHD-IN for Persian Adults.
All the report is solid and powrful, resulting in a good advance in terms of instrumentos for Persian population.
To further improve the article it is advisable to:
Abstract:
1. Stablish the place and location where the scale was used for the validation process
2. Inform the age range, and % of olders, and SD
3. Highlight the usefulness of the instrument
Method
- Inform the recruitment process, site, location, place, institution of participants.
- Inform the % of olds per range of age. The range Is from 18 to 65, but the mean is 23 with SD os 7... then the participants are youngs... is important for understand the validation
- Inform about the rol of original author of the instuments in the process of translation
- Inform about some details of the translation, adaptation or location of the translation of the instrument
Discussions
Is a underutilised section; because it does not really discuss, or reflects in a very superficial way for the potential of article. For example: why 7 items are good in this case?, or why the original 15 items are becoming less to 10 or to 7?
And the limitations: the authors do not mention the characteristics of the sample: high levels of education and young. It is important to mention that this version would work well with this type of population.
Tables
- Use the leyend to explain the acronim "ns"
Cites and References
- review some cite with author names an no numbers (e.g. Becker, 2021 line 193)
- Delete a double doi.org in line 445
- 29% of the reference are from Becker
Other
- Use subindex letter for (df) degrees of freedom of X2 (e.g. see line 272 and 284)
Supplementary material
Please, if you want to help the world, release and link access to the instrument, correction and tables with open access
Author Response

(The authors gave the same response as above.)
